# Successful elimination of falciparum malaria following the introduction of community-based health workers in Eastern Myanmar: A retrospective analysis

**Aye Sandar Zaw**[1,2], **Ei Shwe Sin Win**[1], **Soe Wai Yan**[1], **Kyaw Sithu Thein**[1],
**Vasundhara Verma**[2], **Alistair R. D. McLean**[1,2], **Thar Tun Kyaw**[1,2,3], **Nicholas J. White**[4,5],
**Frank M. Smithuis**[1,2,4,5]*

**1** Medical Action Myanmar, Yangon, Myanmar, **2** Myanmar Oxford Clinical Research Unit, Yangon, Myanmar, **3** Department of Public Health, Ministry of Health and Sports, Nay Pyi Taw, Myanmar, **4** Centre for Tropical Medicine and Global Health, Nuffield Department of Clinical Medicine, University of Oxford, Oxford, United Kingdom, **5** Mahidol-Oxford Tropical Medicine Research Unit (MORU), Faculty of Tropical Medicine, Mahidol University, Bangkok, Thailand

* frank.m.smithuis@gmail.com

**Data Availability Statement:** The data underlying this publication are available freely upon request to the Mahidol-Oxford Tropical Medicine Research

## Abstract

### Background

Myanmar has a large majority of all malaria in the Greater Mekong Subregion. In the past decade, substantial progress was made in malaria control. The residual burden of malaria is in remote areas where currently recommended malaria elimination approaches are generally not feasible. In such hard-to-reach communities in Mon state, East Myanmar, Medical Action Myanmar introduced community health workers (CHWs) to deliver early diagnosis and treatment for malaria. We conducted a retrospective analysis to assess the impact of this intervention.

### Methods and findings

This retrospective analysis involved data collected routinely from a CHW programme in Mon state conducted between 2011 and 2018. A network of 172 CHWs serving a population of 236,340 was deployed. These CHWs carried out 260,201 malaria rapid diagnostic tests (RDTs) to investigate patients with acute febrile illness. The median blood examination rate was 1.33%; interquartile range (IQR) (0.38 to 3.48%); 95% CI [1.28%, 1.36%] per month. The changes in malaria incidence and prevalence in patients presenting with fever were assessed using negative binomial regression mixed effects models fitted to the observed data. The incidence of *Plasmodium falciparum* malaria (including mixed infections) declined by 70%; 95% CI [65%, 75%]; $p < 0.001$ for each year of CHW operation. The incidence of *P. vivax* malaria declined by 56%; 95% CI [50%, 62%]; $p < 0.001$ per year. Malaria RDT positivity rates for *P. falciparum* and *P. vivax* declined by 69%; 95% CI [62%, 75%]; $p < 0.001$ and 53%; 95% CI [47%, 59%]; $p < 0.001$ per year, respectively. Between 2017 and 2018, only 1 imported *P. falciparum* case was detected in 54,961 RDTs. The main limitations of the study

Unit Data Access Committee, complying with the data access policy. Queries and applications for data should be directed to datasharing@tropmedres.ac.

**Funding:** The malaria project was supported by The Global Fund to fight AIDS, Tuberculosis and Malaria and Medical Action Myanmar. Award receiver: FMS Grant number: UNOPS_RAI_MAM_001 https://www.theglobalfund.org/en/ This retrospective analysis has been conducted with the support of the Wellcome Trust. Award receiver: FMS Grant number: 220211/Z/20/Z https://wellcome.org/ This research was funded in whole, or in part, by the Wellcome Trust (Grant number 220211). For the purpose of open access, the author has applied a CC BY public copyright licence to any Author Accepted Manuscript version arising from this submission. The funders had no role in study design, data collection and analysis, decision to publish, or preparation of the manuscript.

**Competing interests:** The authors have declared that no competing interests exist.

**Abbreviations:** ABER, annual blood examination rate; ACD, active case detection; CHW, community health worker; DoH, Department of Health; GFATM, Global Fund to fight AIDS, Tuberculosis, and Malaria; GMS, Greater Mekong Subregion; IQR, interquartile range; LLIN, long-lasting insecticidal net; MAM, Medical Action Myanmar; MBER, monthly blood examination rate; NMCP, National Malaria Control Programme; PCD, passive case detection; RDT, rapid diagnostic test; WHO, World Health Organization.

are use of retrospective data with possible unidentified confounders and uncharacterised population movement.

## Conclusions

The introduction of CHWs providing community-based malaria diagnosis and treatment and basic health care services in remote communities in Mon state was associated with a substantial reduction in malaria. Within 6 years, *P. falciparum* was eliminated and the incidence of *P. vivax* fell markedly.

---

## Author summary

### Why was this study done?

- Malaria is still a major health problem in Myanmar, which has a large majority of all the reported malaria cases in the Greater Mekong Subregion.

- In Myanmar, malaria is prevalent predominantly in remote and difficult to access communities where currently recommended malaria elimination approaches are generally not feasible.

- In 2011, Medical Action Myanmar introduced community health workers (CHWs) to deliver early diagnosis and treatment for malaria in hard-to-reach communities in Mon state, East Myanmar. This retrospective study was performed to assess the impact of this intervention on the incidence and prevalence of malaria.

### What did the researchers do and find?

- We conducted an analysis of the uptake of CHW services and the effectiveness of this programme in controlling and eliminating malaria.

- Overall, 172 CHWs were supported to serve a population of 236,340 people. The CHWs carried out 260,201 malaria rapid diagnostic tests (RDTs) over an 8-year period.

- There was a rapid and sustained reduction in the incidence of both falciparum and vivax malaria.

- Falciparum malaria was eliminated within 6 years, and vivax malaria was reduced markedly.

### What do these findings mean?

- Although this is a retrospective observational analysis, it suggests that elimination of falciparum malaria was achieved simply by supporting CHWs to provide effective early detection and treatment of malaria.

- There was also a substantial reduction in the incidence of vivax malaria, but *P. vivax* was more difficult to eliminate than *P. falciparum*.

- This is a low transmission setting. Additional measures may be required to eliminate malaria where there are foci of higher transmission

- Continued support and broadening of the remit of CHWs to include basic healthcare support is needed to ensure that malaria elimination is sustained.

## Introduction

To counter the threat posed by worsening artemisinin and partner drug resistance in *Plasmodium falciparum*, the countries of the Greater Mekong Subregion (GMS) have agreed to aim for elimination of falciparum malaria by 2025 [1,2]. This ambitious initiative has been supported by generous funding from the Global Fund to fight AIDS, Tuberculosis, and Malaria (GFATM). In the GMS, malaria is now prevalent predominantly in remote and difficult to access communities mainly in Myanmar, which has a large majority of all the reported malaria cases in the region [2,3].

The World Health Organization (WHO) has developed several detailed "malaria elimination manuals" in recent years [1–5]. These all recommend a broad range of labour-intensive and costly activities. Initially, the activities were divided into 2 phases. The first was a "transmission-reduction phase," which involved aggressive scale-up of effective preventive and curative interventions to achieve universal coverage. This phase aimed to reduce the malaria incidence to a level at which elimination could be considered (below 1 case per 1,000 people at risk per year). This was then followed by an "elimination phase," which aimed to reduce malaria incidence to zero. In a recent update from WHO, these thresholds have been removed [4], but the recommended intensive elimination activities have been retained. Malaria case detection and entomological surveillance are the core interventions during elimination. It is recommended that every *P. falciparum* malaria case should be investigated and managed to avoid onward transmission. Based on the foci of transmission identified, appropriate vector control and antimalarial drug-based interventions (reactive case detection) should be deployed to interrupt transmission rapidly [4]. Recently, the WHO Mekong Malaria Elimination Programme has advised adding weekly active fever screening of all target communities in the elimination phase [5].

This extensive (and expensive) package of malaria elimination activities currently recommended by WHO is not supported by evidence [6]. It is also challenging, labour intensive, and it needs substantial and sustained financial resources, which must be continued well after malaria is eliminated. Is this elaborate process honestly feasible in the remote areas of Myanmar where most of the GMS malaria now persists? If this advice is followed, then qualified staff must travel frequently to remote areas with little or no infrastructure to investigate malaria cases. But it is often impossible to reach remote communities quickly (or at all) during the rainy season when malaria transmission is highest. Entomological surveillance and reactive case detection require trained mobile teams. The activities recommended by WHO are difficult, demanding, very time consuming, and unlikely to yield reliable results. With an uncertain financial future for regional malaria control activities, it is important to identify the core malaria control elements that must be sustained to complete elimination [6].

Provision of long-lasting insecticidal nets (LLINs) and universal coverage of early diagnosis and effective treatment of clinical malaria infections are generally accepted as a first priority of malaria management [6–8]. In Myanmar, as in many malaria-endemic countries, the remote rural communities have very limited access to trained health staff. To pursue the National goal

of malaria control and elimination and so to provide these remote communities with ready access to early diagnosis and treatment, a large number of community health workers (CHWs) were recruited from the rural population and trained over the past decade. Medical Action Myanmar (MAM), a medical organization, introduced and supported a network of 172 CHWs in remote areas of Mon state (Southeast Myanmar) between 2011 and 2018, after which the majority of the program activities were handed over to the Department of Health (DoH) on request of the National Malaria Control Programme (NMCP). We conducted a retrospective analysis of routinely collected data from this CHW programme to assess the uptake of CHW services and the effectiveness of this programme in controlling and eliminating malaria.

## Methods

This is a retrospective analysis of existing datasets obtained from a CHW programme in Mon state, Myanmar. The details of the programme, data collection methods, and statistical analyses are described below. Standard malariometric data were analysed. There were no data-driven changes to the analyses. This study is reported as per the Strengthening the Reporting of Observational Studies in Epidemiology (STROBE) guideline (S1 Checklist).

### Area and population

The project was carried out in 4 "townships" (Kyaikmaraw, Mudon, Paung, and Ye) of the Mon state. The terrain varies from flat coastal areas in the West, bordering the Andaman Sea, to more hilly forested areas in the East. Mon state has a tropical monsoon climate with a hot and dry summer season from March to April, rainy season from May to October, and cooler season from November to February. Yearly rainfall is approximately 5,000 mm/year. Malaria transmission intensity is low to moderate and seasonal (i.e., most malaria is during or shortly after the rainy season). Many villages are difficult to access, particularly during the rainy season.

### Community health worker (CHW) recruitment, training, and deployment

CHWs are community members who volunteered to conduct healthcare activities within their community. MAM introduced CHWs in remote communities in Mon state in consultation with the Myanmar NMCP. The CHWs were trained to use malaria rapid diagnostic tests (RDTs) and to screen all villagers presenting with fever and, if the malaria RDTs were positive, to treat them with appropriate antimalarial drugs (see below). CHWs continued with their usual farming or household activities but agreed to be available for consultation for ≥1 hour every day, in the early morning or late afternoon, when most villagers returned from their field work. The CHWs received a fixed monthly incentive of 5,000 kyats (then equivalent to 4 USD), plus 500 kyats for every patient tested.

The first CHWs were introduced in April 2011. Thereafter, the network gradually expanded to 172 CHWs across the 4 townships (large areas) in Mon state. The project continued until the end of 2018 after the majority of the activities were handed over to the NMCP. During the project period, the Myanmar DoH introduced basic healthcare staff in some project communities. In those cases, these DOH staff took over the responsibility for malaria management, and these villages were excluded subsequently from the project and, thus, this evaluation.

MAM mobile medical teams visited each community each month for supervision, supply, and continued training of CHWs. Routine data were collected, including basic patient information details, test results, and the treatments provided. Information about the duration of fever before the RDT was collected since 2013 only. All data were recorded by the CHWs

using standard forms, later entered in a data file, and copies were submitted to the NMCP office each month.

## Malaria diagnosis and treatment

Routine passive case detection (PCD; i.e., testing of any patient with complaints of fever) using a malaria RDT was undertaken by CHWs. The RDT kits used were SD Bioline Malaria RDTs (Alere/SD, South Korea). From 2011 to 2014, these diagnosed *P. falciparum* specifically (*Pf*HRP2 detection) and also had a pan malaria antigen detection band, so *P. vivax* and other malarias could be inferred from pan antigen test positivity but negativity in the *Pf*HRP2 band. Such cases were all assumed to be *P. vivax* malaria. After 2014, the newer tests identified *P. vivax* specifically, so mixed infections could be speciated. In line with national guidelines, *P. falciparum* malaria was treated with the standard oral artemether–lumefantrine regimen (artemether 20 mg–lumefantrine 120 mg coformulated tablets according to body weight provided in colour-coded blister packages and taken 2 times per day for 3 days). To reduce transmissibility of *P. falciparum*, a single dose of primaquine (0.75 mg base/kg according to the National recommendation) was given on the first day of treatment except in pregnant and breastfeeding women, and infants under the age of 1. *P. vivax* malaria was treated with 3 days of chloroquine (total dose 25 mg base/kg) plus primaquine (target dose 0.75 mg base/kg), which was given without G6PD testing, once weekly for 8 weeks according to the National recommendation, in order to mitigate the risk of haemolysis in people with G6PD deficiency. The weekly doses were 15 mg, 30 mg, and 45 mg of primaquine for 5 to 9 year olds, 10 to 14 year olds, and 15 year and older patients, respectively. Patients with severe malaria were referred to the nearest hospital.

## Integrated basic healthcare services delivered by CHWs

Within 2 years of starting the programme, it became clear that malaria incidence had decreased substantially, and other causes of febrile illness were dominating consultations. An integrated basic healthcare package was then introduced to address this changing pattern of needs, while support continued for early detection and treatment of malaria. This comprised 4 components:

1. Management of selected common diseases such as respiratory tract infections, diarrhoea, and skin infections.

2. Detection of acute malnutrition using mid-upper arm circumference measurement and subsequent treatment

3. Active case finding of tuberculosis and referral to the nearest government hospital.

4. Referral of complicated and severely ill patients to the nearest government hospital.

CHWs received training in line with the curricula of the NMCP and the National TB programme. All referral costs were provided by MAM.

## Additional interventions

In collaboration with the NMCP, the CHWs, and MAM mobile medical teams also engaged in mass distribution of LLINs during the first 4 years of the programme. Additional LLINs were made available for newcomers to the village, migrants, and pregnant women. CHWs and MAM mobile medical teams also organized health education and community engagement discussions to inform the community and to seek their feedback.

### Ethics approval

This analysis of retrospective observational data was exempt from ethical review by the Oxford Tropical Research Ethics Committee (OxTREC).

### Statistical analysis

The objective of the analysis was to describe the temporal changes in malaria (both falciparum and vivax) that followed implementation of the CHW programme. Seasonality and geography are important covariates that are normally included in malariometric evaluations. RDT positivity rates and monthly blood examination rates were calculated from monthly malaria reports. Negative binomial mixed effects regression models were constructed with random intercepts and slopes to assess temporal changes in incidence and RDT positivity rates. Seasonal and geographic variations in outcomes were accounted for in the multivariate models.

When modelling incidence and RDT positivity rates, the outcome was malaria (RDT-positive) results (per month, per CHW) and the primary exposure was years of CHW operation. The exposure variable for the incidence model was the population served by a CHW, and the exposure variable for the RDT positivity rate model was the number of RDTs performed. When calculating annual blood examination rates (ABERs) and monthly blood examination rates (MBERs), the population each CHW served was incorporated as the exposure variable and the outcome variable was the total RDTs performed. The exposure variable is, therefore, RDTs performed on patients who got the care of a CHW for their fever. The uptake of CHW services was high throughout the study period, and it is very likely that the majority of people with fever sought the free healthcare services provided by the CHW. We assume that the RDT positivity rate in these people is similar to that in the minority who sought care elsewhere.

CHWs were introduced on different days of the month so the population for the first month of operation was calculated taking into account the starting date of the CHWs. When the exact starting dates were unknown, we chose the starting date in the middle of the month. For the CHWs in Kyaikmaraw, Mudon, and Paung townships, the project activities stopped at the end of 2018, and the last data were collected in December. The exact last day of the project sites was not known, so we assumed the project end date was the middle of December 2018 and the population covered was calculated accordingly.

Statistical analysis was done using STATA version 15 (StataCorp, College Station, Texas, United States).

## Results

### Recruitment of CHWs and population coverage

The first 12 CHWs were recruited in Paung township in April 2011, followed by recruitment of 27 CHWs in Kyaikmaraw and 14 CHWs in Mudon townships in May, and 28 CHWs in Ye township in September 2011. Over the observed period, the CHW network gradually expanded to 172 communities serving a total population of 236,340. This was approximately 27.2% of the population of these 4 townships and 11.5% of the population of Mon state [9]. The populations covered by individual CHWs were generally small; with a median (interquartile range (IQR)) of 778 (394 to 1,752) people.

### Heterogeneity in baseline malaria incidence and RDT positivity rates across the region

At the beginning of CHW operations (i.e., the first year of CHW operation), there was marked heterogeneity in the baseline malaria incidence and RDT positivity rates between different townships (Table 1).

**Table 1. Distribution of malaria in each township at the first year of CHW operation.**

| Township (Number of villages covered) | Average population covered | Number of RDT performed | MBER (per 100 population) | *P. falciparum + mixed infections* | | | *P. vivax* | | |
|---|---|---|---|---|---|---|---|---|---|
| | | | | Number of cases | Incidence[a] | Positivity rate[b] | Number of cases | Incidence[a] | Positivity rate[b] |
| Kyaikmaraw (75) | 102,025 | 27,471 | 2.24 | 438 | 4.29 | 1.59 | 502 | 4.92 | 1.83 |
| Mudon (14) | 32,028 | 4,727 | 1.23 | 44 | 1.37 | 0.93 | 83 | 2.59 | 1.76 |
| Paung (22) | 36,619 | 8,015 | 1.82 | 402 | 10.98 | 5.02 | 324 | 8.85 | 4.04 |
| Ye (61) | 29,420 | 11,517 | 3.26 | 279 | 9.48 | 2.42 | 343 | 11.66 | 2.98 |

[a]Per 1,000 person-years.

[b]Per 100 RDTs.

CHW, community health worker; MBER, monthly blood examination rate; RDT, rapid diagnostic test.

## Malaria rapid diagnostic testing

Malaria transmission was seasonal with a peak in cases at the start of the rainy season and a smaller peak shortly after the rainy season (Fig 1 and S1 Table). Between 2011 and 2018, 260,201 RDTs were performed of which 4,648 (1.79%) were positive for malaria; 1,789 patients

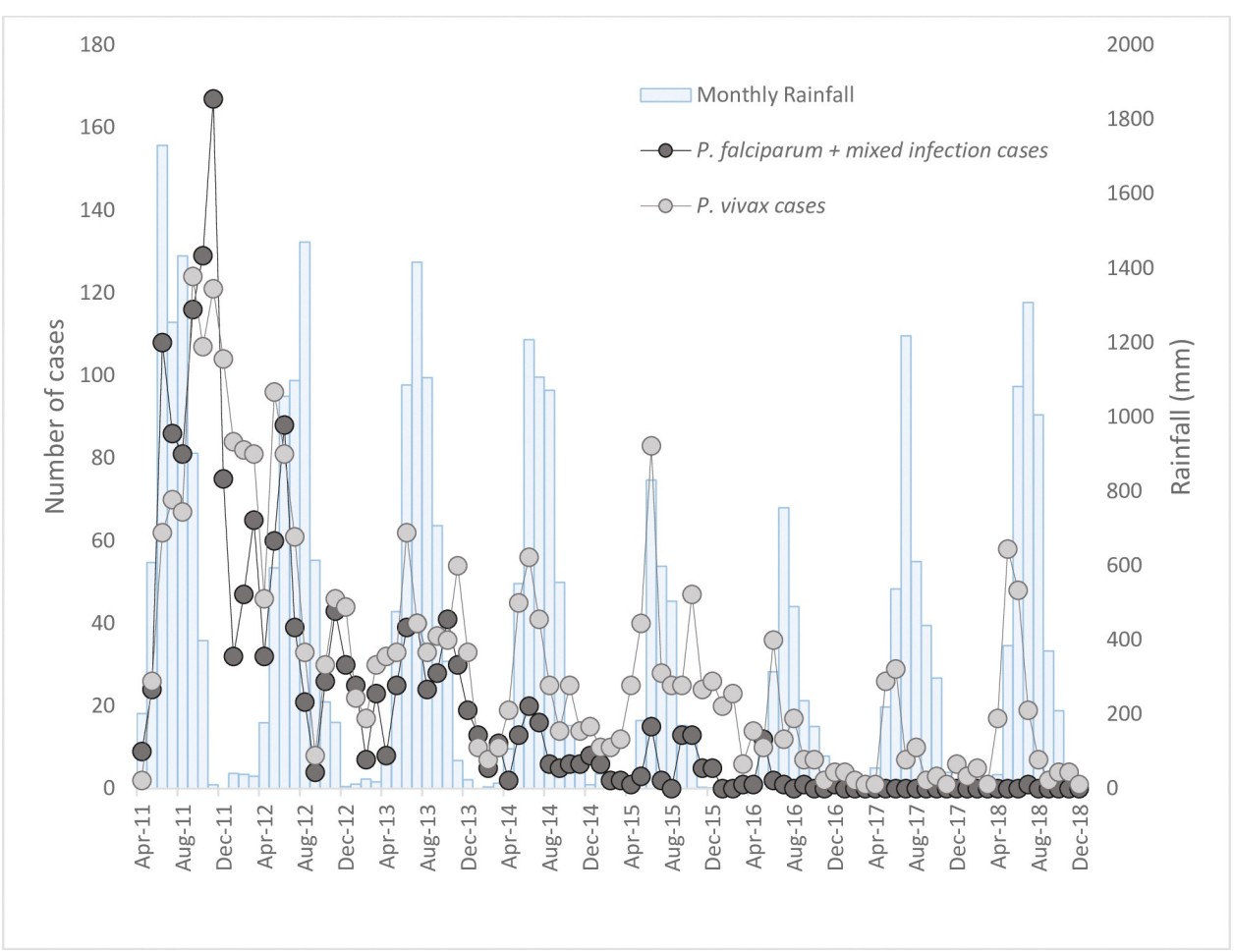

**Fig 1. Number of malaria cases and monthly rainfall [10] in Mon state between 2011 and 2018.**

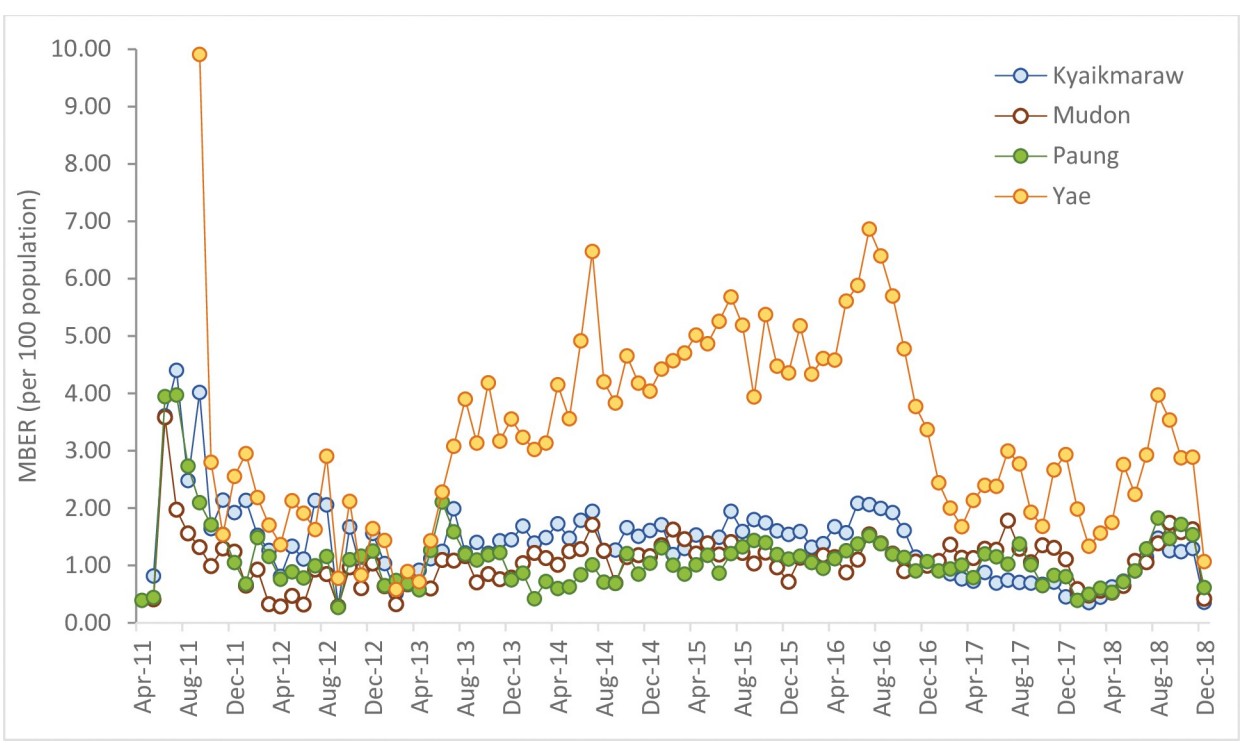

**Fig 2. Trend of monthly blood examination rate (MBER) in 4 townships between 2011 and 2018.**

were positive for *P. falciparum* (includes mixed infections), and 2,859 were positive for *P. vivax* infection. Six patients were referred to hospital for further management: 2 from Mudon, 1 from Paung, and 3 from Ye. There were no cases of blackwater fever and no malaria deaths reported during the entire project.

Since 2013, information about the onset of fever was collected. In 1,246 of 1,868 (66.7%) malaria patients, the onset of fever was within 24 hours of their visit to the CHW.

The median MBER across all CHWs over the observed period was 1.33%; (IQR 0.38% to 3.48%); 95% CI [1.28%, 1.36%] per month. The highest median MBER (2.22%) was observed in Ye township, the township with the highest baseline malaria incidence, and the most dense forest coverage. The lowest median MBER (0.89%) was in Mudon township, the township with the lowest baseline malaria incidence and the least dense forest coverage (Fig 2).

## Reduction in malaria incidence with elimination of *P. falciparum*

There was an exponential decline in malaria incidence and RDT positivity rates for both *P. falciparum* and *P. vivax* following the introduction of CHWs, until *P. falciparum* was eventually eliminated (Figs 3, 4, 5 and 6). The *P. falciparum* (+mixed infection) incidence fell from 7.90 cases per 1,000 person-years in 2011 to 0.005 in 2018, which is a yearly reduction of 70%; 95% CI [65%, 75%] per year of CHW operation (Fig 7 and Tables 2 and 3). This equates to a halving time of about 6 months. The RDT positivity rates for *P. falciparum* fell from 2.80% to 0.004% in the same period, a 69%; 95% CI [62%, 75%] yearly reduction (Fig 8 and Tables 2 and 3). During the last 2 years of the project period, only a single imported positive case of *P. falciparum* was detected out of 54,961 patients tested. This person had been stationed on the border with the Karen state and was tested in Mon state while on home leave. Thus, in terms of endogenous cases, *P. falciparum* malaria was eliminated.

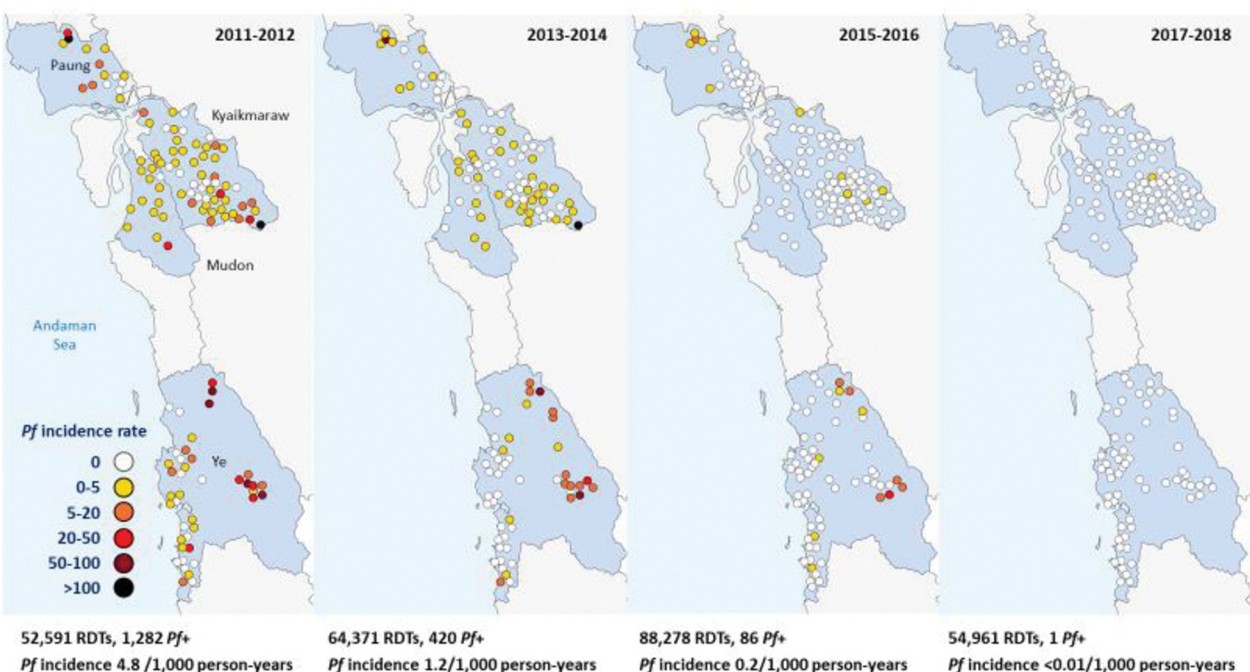

**Fig 3. Incidence rate of *P. falciparum* malaria (including mixed infections) following the introduction of CHWs between 2011 and 2018.**

## *P. vivax* dominates as *P. falciparum* approaches elimination

The decline in *P. vivax* incidence was also substantial, but less than for *P. falciparum*. *P. vivax* incidence fell from 6.78 to 0.83 cases per 1,000 person-years. The corresponding decline in *P.*

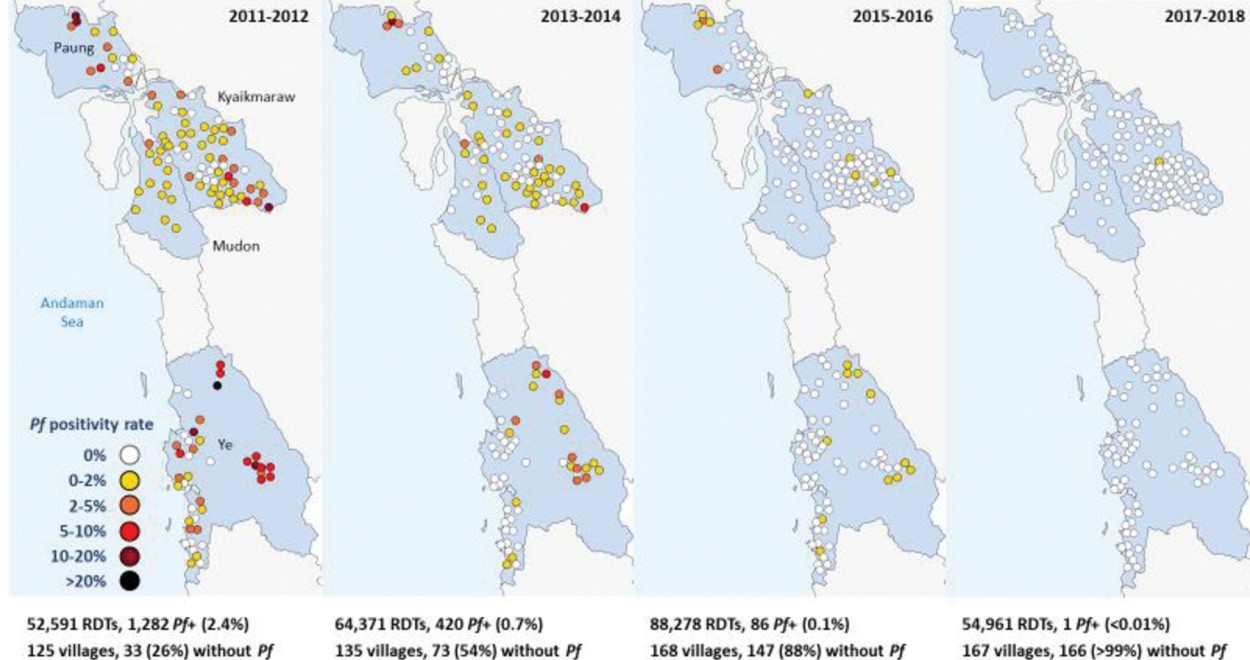

**Fig 4. RDT positivity rate of *P. falciparum* malaria (including mixed infections) following the introduction of CHWs between 2011 and 2018.**

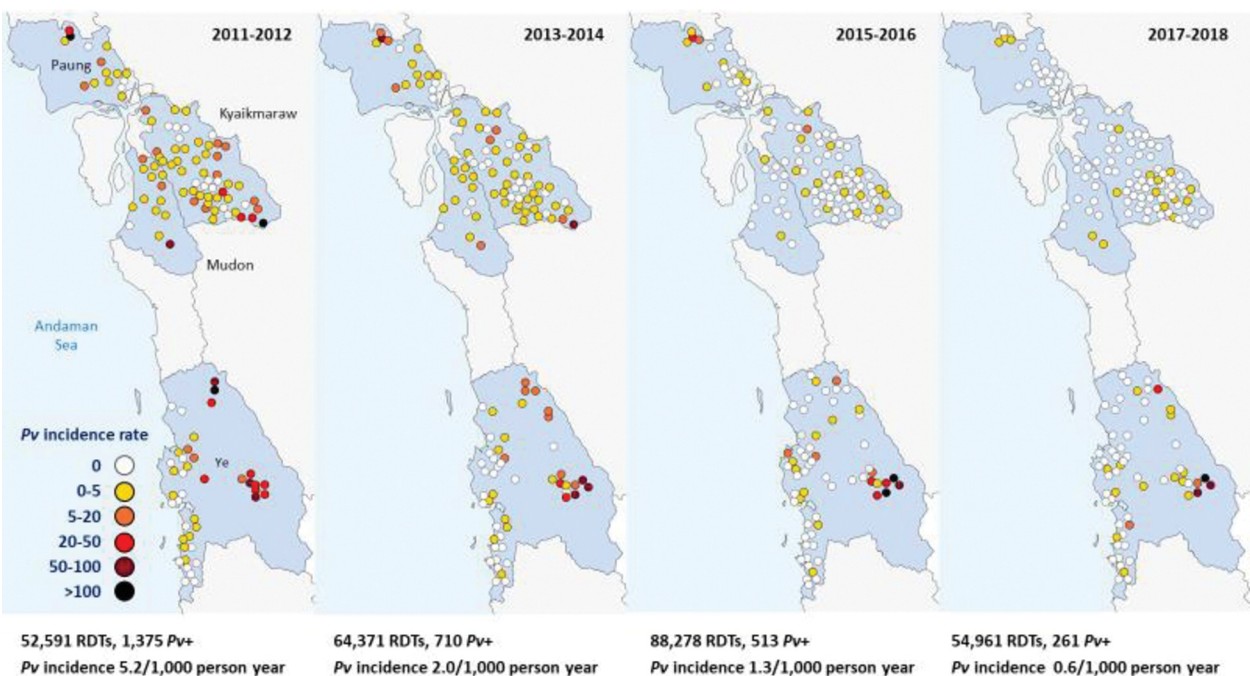

**Fig 5. Incidence rate of *P. vivax* malaria following the introduction of CHWs between 2011 and 2018.**

*vivax* positivity rates was from 2.41% to 0.61% (Table 2). This equates to a 56%; 95% CI [50%, 62%] reduction in *P. vivax* incidence and a 53%; 95% CI [47%, 59%] reduction in *P. vivax* positivity rate per year of CHW operation (Figs 9 and 10 and Table 3).

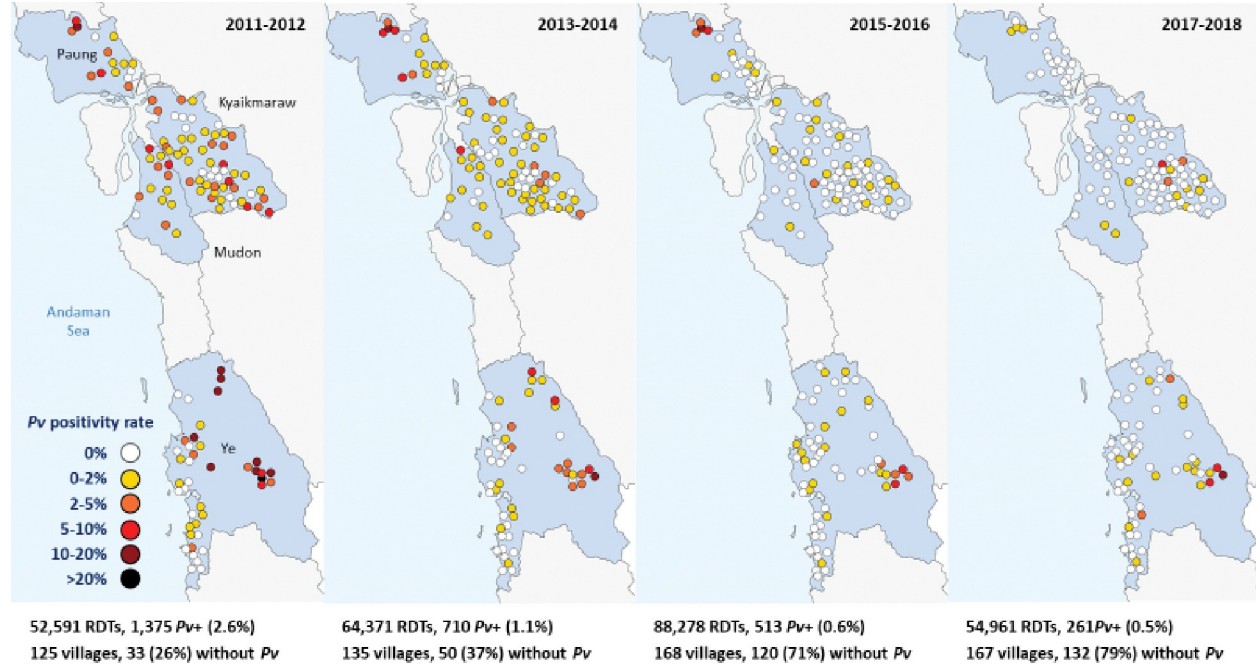

**Fig 6. RDT positivity rate of *P. vivax* malaria following the introduction of CHWs between 2011 and 2018.**

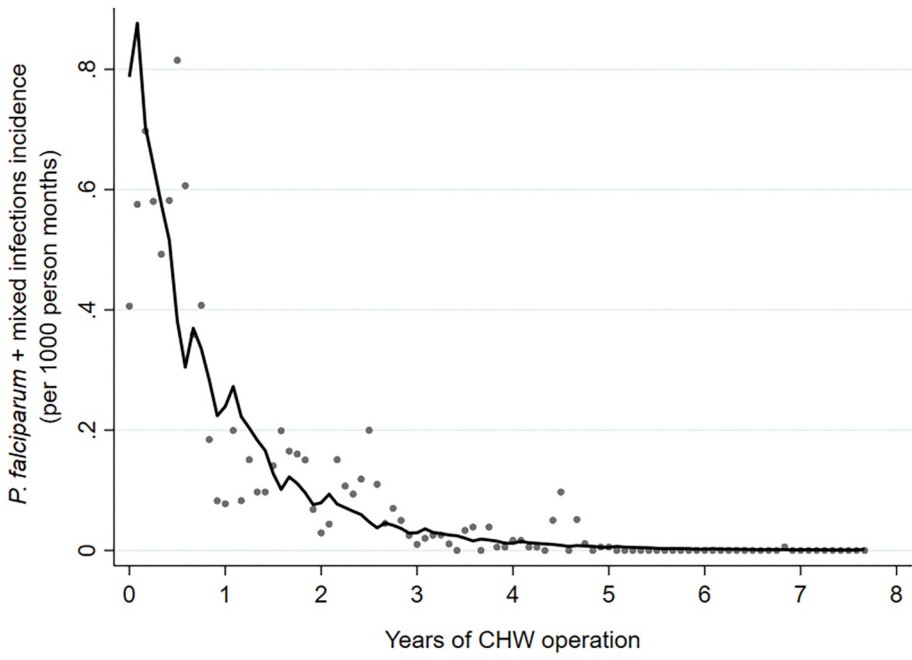

**Fig 7. Incidence rate of *P. falciparum* malaria (including mixed infections) by years of CHW operation.** Grey dots represent observed data, and the line indicates the prediction from a mixed effects negative binomial regression model.

At the beginning of the observation period, the incidence of *P. falciparum* (+mixed infections) was higher than that of *P. vivax* (7.90 vs 6.78 cases per 1000 person-years; a ratio of 1.16). The pattern then reversed with the preponderance of *P. vivax* becoming more pronounced over time. (P<0.001).

## Case detection

In addition to the PCD performed by the CHWs, MAM mobile teams performed active case detection (ACD), including school surveys, in 2011, 2012, and 2013 in 22, 39, and 48 villages,

**Table 2. Malaria incidence and RDT positivity rate after introduction of CHWs by calendar year.**

| Year | Number of villages[a] | Person years | Number of RDTs performed | MBER per 100 population | *P. falciparum* (+ mixed infections) | | | *P. vivax* | | | *P. falciparum/ P. vivax* ratio |
|------|------|------|------|------|------|------|------|------|------|------|------|
| | | | | | Number of cases | Incidence[b] | Positivity rate[c] | Number of cases | Incidence[b] | Positivity rate[c] | |
| **2011** | 114 | 100,688 | 28,346 | 2.35 | 795 | 7.90 | 2.80 | 683 | 6.78 | 2.41 | 1.16 |
| **2012** | 124 | 164,567 | 24,245 | 1.23 | 487 | 2.96 | 2.01 | 692 | 4.20 | 2.85 | 0.70 |
| **2013** | 132 | 177,883 | 27,079 | 1.27 | 309 | 1.74 | 1.14 | 429 | 2.41 | 1.58 | 0.72 |
| **2014** | 133 | 182,464 | 37,292 | 1.70 | 111 | 0.61 | 0.30 | 281 | 1.54 | 0.75 | 0.40 |
| **2015** | 162 | 184,968 | 41,701 | 1.88 | 67 | 0.36 | 0.16 | 355 | 1.92 | 0.85 | 0.19 |
| **2016** | 168 | 198,446 | 46,577 | 1.96 | 19 | 0.10 | 0.04 | 158 | 0.80 | 0.34 | 0.12 |
| **2017** | 166 | 209,731 | 27,077 | 1.08 | 0 | 0 | 0.00 | 92 | 0.44 | 0.34 | 0.00 |
| **2018** | 167 | 202,699 | 27,884 | 1.15 | 1 | 0.005 | 0.004 | 169 | 0.83 | 0.61 | 0.006 |

[a]The number of villages under implementation within the year.

[b]Per 1,000 person-years.

[c]Per 100 RDTs.

CHW, community health worker; MBER, monthly blood examination rate; RDT, rapid diagnostic test.

**Table 3. Reduction (%) in incidence and RDT positivity rate/year of CHW operation between 2011 and 2018.**

| | Incidence rate ratio per year of CHW operation* (95% CI) | Percentage reduction per year of CHW operation (95% CI) | P value** |
|---|---|---|---|
| *P. falciparum + mixed infections* | | | |
| - Incidence | 0.30 (0.25–0.35) | 70 (65–75) | <0.001 |
| - RDT positivity rate | 0.31 (0.25–0.38) | 69 (62–75) | <0.001 |
| *P. vivax* | | | |
| - Incidence | 0.44 (0.38–0.50) | 56 (50–62) | <0.001 |
| - RDT positivity rate | 0.47 (0.41–0.53) | 53 (47–59) | <0.001 |

CHW, community health worker; RDT, rapid diagnostic test; 95% CI, 95% confidence interval.

*Calculated using multivariate model adjusted for seasonal and geographical variation.

**P value from Wald test.

respectively. This activity was added in an effort to reduce malaria rapidly and was focused on villages believed beforehand to have a high malaria burden based on the RDT positivity rate reported by the CHWs. In 2015, small-scale ACD was conducted in 2 locations for migrant people, and in 2018 in 5 locations as part of a new "elimination phase" policy of the NMCP to conduct reactive case detection in communities where new cases of malaria were identified.

Out of 23,495 malaria RDTs tested, 87 (0.37%) *P. falciparum* (+ mixed) cases and 62 (0.26%) *P. vivax* cases were identified and treated. The majority of cases came from one village in Paung township, where the CHW had reported the highest RDT positivity rate for malaria (35%) of the entire project in 2011. Out of 1,174 persons tested in that village by ACD, 74

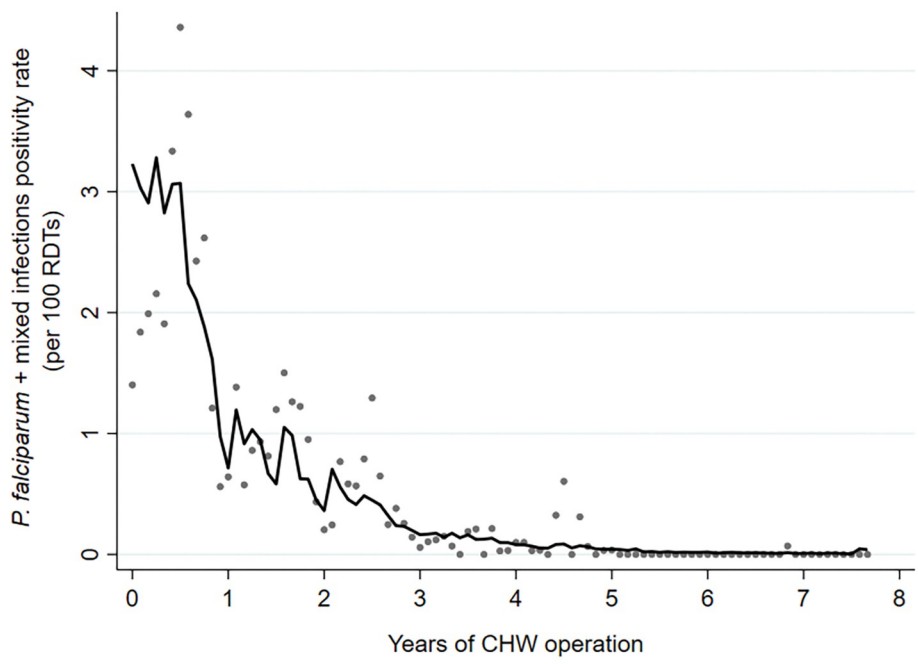

**Fig 8. RDT positivity rate of *P. falciparum* malaria (including mixed infections) by years of CHW operation.** Grey dots represent observed data, and the line indicates the prediction from a mixed effects negative binomial regression model.

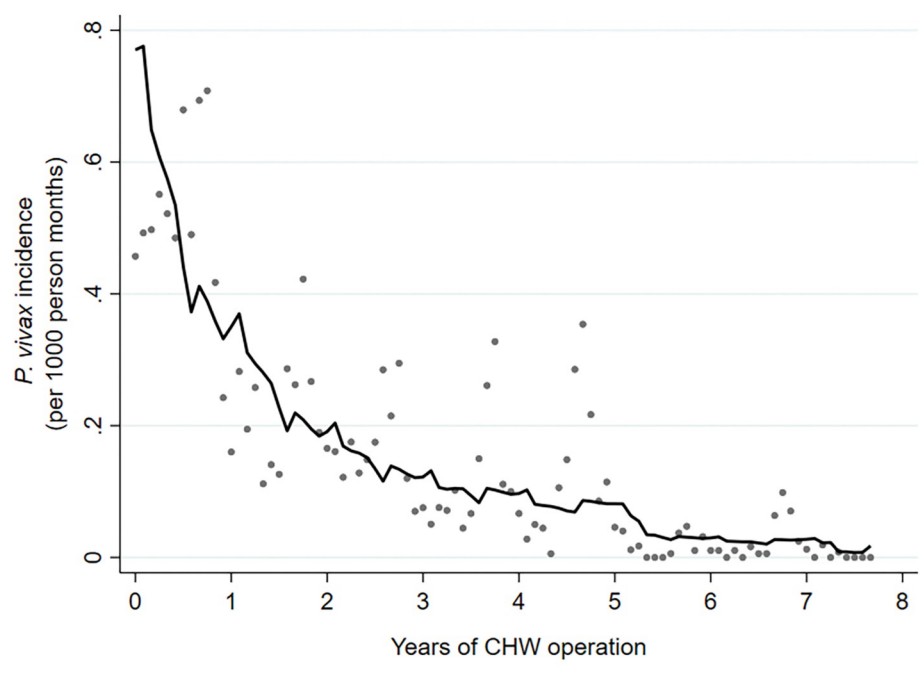

**Fig 9. Incidence rate of *P. vivax* malaria by years of CHW operation.** Grey dots represent observed data, and the line indicates the prediction from a mixed effects negative binomial regression model.

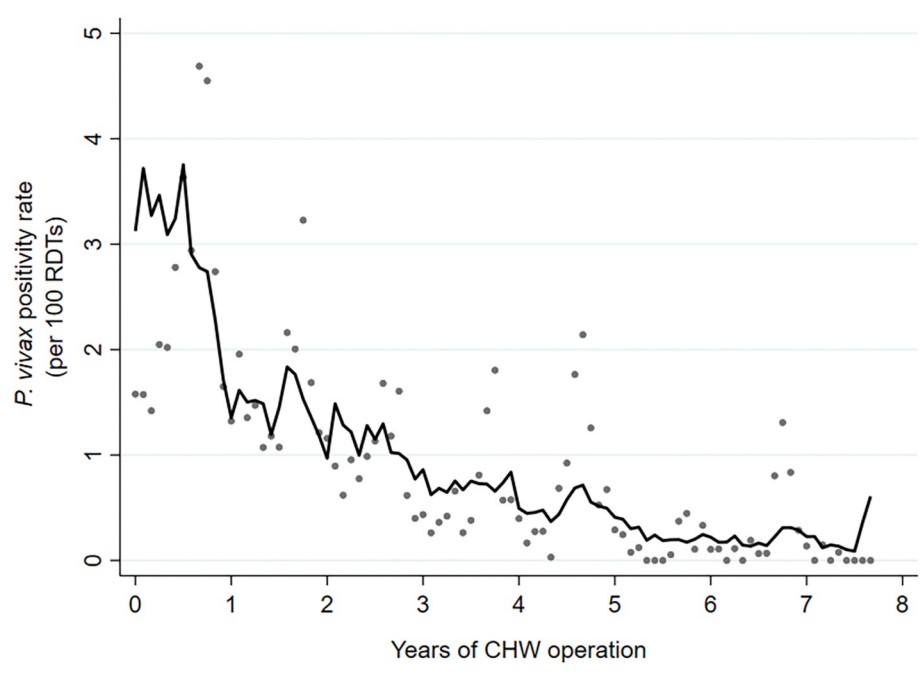

**Fig 10. RDT positivity rate of *P. vivax* malaria by years of CHW operation.** Grey dots represent observed data, and the line indicates the prediction from a mixed effects negative binomial regression model.

(6.3%) tested positive for *P. falciparum* and 53 (4.5%) for *P. vivax*. Of the remaining villages, the yield was very low, with 22,321 persons tested and 13 *P. falciparum* (0.06%) and 9 *P. vivax* (0.04%) infections identified. The majority of ACD-positive cases were identified in 2011 (S2 Table).

## LLIN distribution

A total of 57,684 LLINs were distributed in the project communities. Distribution was conducted according to the assignment by NMCP; 48,646 LLINs were provided through mass distribution from 2011 to 2014, and 9,038 LLINs were provided as required by CHWs to newcomers, migrants, and pregnant women.

## Discussion

In remote villages of Mon state in Eastern Myanmar, where malaria transmission was low, falciparum malaria was eliminated within 6 years, and vivax malaria was reduced substantially following the introduction of and support for village-based CHWs. Over the last 2 years of the observation period, only 1 case of falciparum malaria was identified out of 54,961 tests performed, and that person had imported the infection, emphasizing the importance of continued vigilance even after elimination goals have been achieved. Elimination of falciparum malaria was achieved simply by supporting CHWs to provide effective early detection and treatment of malaria.

These remote communities in Mon state always had limited access to quality malaria diagnosis and treatment [11]. Clinics are far away from most villages and transport is difficult and expensive. Before the introduction of CHWs, the villagers had access only to untrained informal health care providers who did not have malaria tests. Without a diagnosis for undifferentiated fever, a variety of inappropriate treatments were taken and malaria transmission continued unabated. As a consequence, there were no reliable historical malariometric data from the studied areas. As malaria transmission is of low to moderate intensity in this region, and there have been no major climatic or environmental changes then, if the village-based CHWs had not been deployed, it is likely that transmission would have continued unchanged. After the introduction of CHWs, the communities had rapid access to an accurate diagnosis and free treatment for malaria and reliable malariometric data were now collected regularly and systematically. The programme uptake was high because the services were in the village and were free of charge.

Why was the fall in falciparum malaria so rapid? Early treatment with ACT plus single dose primaquine promptly decreased gametocytaemia and, thus, transmission [12]. As malaria was reduced, the proportion of asymptomatic carriers presumably decreased [13–15]. Declining immunity may also have resulted in an increasing proportion of infections being symptomatic and, therefore, receiving effective diagnosis and treatment. The distribution of long-lasting insecticide-treated bed nets (LLINs) probably also contributed to the reduction in malaria, but these are less effective in this region than elsewhere [16] because the main anopheline vectors are exophilic and usually bite in the early evening or early morning [17]. We cannot exclude self-treatment of malaria, but there is no reason to believe either that this was common or that it changed during the period of observation. Besides the introduction of CHWs with RDTs, antimalarials, and LLINs, there were no other substantial antimalaria interventions to explain the rapid decline in malaria. There is a "tipping point" in the level of malaria transmission below which sustained deployment of diagnosis and effective treatment leads eventually to elimination of malaria. In Mon state, the preceding low level of seasonal transmission meant that this tipping point was readily reached with this single intervention.

As expected, the incidence of *P. vivax* malaria declined more slowly than that of *P. falciparum* [7,15,18]. *P. vivax* is generally more transmissible from the acute infection as infectious *P. vivax* gametocytes are produced synchronously with the asexual parasites [19]. Prompt antimalarial treatment, therefore, limits but does not prevent transmission altogether. In contrast, in acute *P. falciparum* infections, there is a delay in producing transmissible gametocyte densities, and single-dose primaquine and artemisinin derivatives are both potent gametocytocides. The impact of providing rapidly effective treatment is, therefore, greater. The second important factor contributing to the slower decline in *P. vivax* is relapse [20]. Although radical cure with primaquine was provided, the effectiveness of the once weekly primaquine regimen has not been established, and adherence was not monitored. Primaquine causes oxidant haemolysis in G6PD deficiency. The most common G6PD deficiency variant in the Mon is G6PD Mahidol with an allele frequency of approximately 12% [21]. It is notable that there were no reports of severe haemolytic anaemia or blackwater with the once weekly 0.75 mg/kg primaquine regimen. If adherence to the weekly primaquine regimen was poor, relapses following treated infections may well have contributed to continued *P. vivax* transmission [19]. Alternatively, a significant asymptomatic reservoir may have remained [13–15]. Nevertheless, the reduction in vivax malaria was still substantial, suggesting that adherence was good and the once weekly regimen was generally effective.

The high uptake of the services provided by the CHWs shows that their service was appreciated by the population. The uptake decreased during the first 2 years of deployment but then gradually increased after April 2013. Increased uptake of CHW services coincided with the introduction of the broader package of basic health services. Similar observations have been made previously [22]. A "malaria only" service declines in popularity as the proportion of malaria cases falls. People are naturally reluctant to consult services that are increasingly unlikely to help them. However, when transmission is low, a continued high uptake of services is crucial, as every person with fever needs to be tested for malaria if the goal of elimination is to be attained. Improved control of vivax malaria and elimination of falciparum malaria was achieved and sustained by providing other important health care services, including management of other causes of fever. This maintained community interest in seeking healthcare from the CHW.

The main malaria elimination strategy that the GMS countries have been encouraged by WHO to adopt is one of active foci investigation [23,24], notably the 1-3-7 approach. This is very difficult to do properly in remote areas of the GMS. It is expensive and labour intensive, and it is highly inefficient [25]. Intermittent ACD conducted by MAM field staff in low-transmission communities in Mon state yielded extremely low RDT positivity rates that were of no value in directing control activities. The 1-3-7 approach was part of the successful elimination of malaria in China [6,26], but China had substantial resources and different malaria epidemiology and geography. Where resources are limited, diversion of precious resources to an inefficient and unproven strategy will compromise the objective of eliminating malaria [6].

Having achieved malaria elimination, how then can it be sustained? Continued surveillance and effective strategies for the prevention of malaria reemergence in these regions are necessary. Continued support for the CHWs is essential. As shown by McLean and colleagues [22], the introduction of a basic healthcare package, including management of common diseases such as respiratory tract infections and diarrhoea, to the service provided by CHWs resulted in a sustained uptake of malaria testing as malaria elimination was approached. It continued the CHWs' active role while providing interventions relevant to the needs of the community. This sustained CHW popularity and so enhanced their ability to continue to test for malaria and detect reemerging malaria. This integrated care strategy has been shown to be effective in several countries [19,27,28]. Ensuring the quality of CHW service and providing logistic support

is demanding, but it is not as difficult, costly, or complex as the currently recommended malaria elimination strategies [1–5].

This report has several limitations. It was a retrospective observational analysis that limits causal inference. While there was not an obvious alternative explanation for the excellent results, there could be confounders that we have not identified. Patients were not followed up routinely after antimalarial treatment was given so effectiveness was not assessed, although the rapid and substantial impact suggests that the antimalarial drugs were highly effective. Adherence to the 8 weekly primaquine radical cure regimen for vivax malaria cannot be assured, and the continued presence of *P. vivax*, albeit at substantially lower incidence, may be related to this. The success of this simple malaria elimination intervention is attributed to the prevailing low seasonal malaria transmission, but a formal malariometric assessment of transmission intensity and, thus, force of infection was not made. Monitoring the movement of populations was also not possible. Migration is common in this part of Myanmar and poses a substantial challenge to achieving malaria elimination [29,30]. Finally, these observations were made during a more peaceful period. There have been substantial disruptions to health service delivery in recent years related to internal conflict, so it is very uncertain whether these remarkable gains in malaria control and elimination have been sustained.

## Conclusions

Deployment, continued support, and broadening of the remit of CHWs were associated with the elimination of falciparum malaria within 6 years in remote communities living in an area of low seasonal malaria transmission in Mon state, Eastern Myanmar. Falciparum malaria elimination was achieved without deployment of the unproven, labour-intensive, and inefficient reactive case detection and foci investigation strategy, recommended currently by WHO. Continued support for CHWs is needed to ensure that malaria elimination is sustained.

## Supporting information

**S1 Checklist. Strengthening the Reporting of Observational Studies in Epidemiology (STROBE) checklist.**
(DOCX)

**S1 Table. Incidence and RDT positivity rate ratios in the rainy and cool seasons as compared to the summer.**
(DOCX)

**S2 Table. Active case detection.**
(DOCX)

**S1 Fig. Incidence rate of *P. falciparum* malaria (including mixed infections) by years of CHW operation.** Grey dots represent observed data, the grey capped bars show the 95% CIs of each month of observed data in isolation, and the black line indicates the prediction from a mixed effects negative binomial regression model.
(TIF)

**S2 Fig. RDT positivity rate of *P. falciparum* malaria (including mixed infections) by years of CHW operation.** Grey dots represent observed data, the grey capped bars show the 95% CIs of each month of observed data in isolation, and the black line indicates the prediction from a mixed effects negative binomial regression model.
(TIF)

**S3 Fig. Incidence rate of *P. vivax* malaria by years of CHW operation.** Grey dots represent observed data, the grey capped bars show the 95% CIs of each month of observed data in isolation, and the black line indicates the prediction from a mixed effects negative binomial regression model.
(TIF)

**S4 Fig. RDT positivity rate of *P. vivax* malaria by years of CHW operation.** Grey dots represent observed data, the grey capped bars show the 95% CIs of each month of observed data in isolation, and the black line indicates the prediction from a mixed effects negative binomial regression model.
(TIF)

## Acknowledgments

We thank all community health workers in remote communities who were essential for the success of the programme. We thank all MAM field staff for their commitment and hard work under very difficult circumstances in remote communities and the MAM data team for data curation. We also acknowledge the National Malaria Control Programme and staff from United Nations Office for Project Services in Myanmar for their support. MOCRU is part of the Mahidol Oxford Tropical Medicine Research Programme.

## Author Contributions

**Conceptualization:** Nicholas J. White, Frank M. Smithuis.

**Data curation:** Aye Sandar Zaw, Ei Shwe Sin Win, Kyaw Sithu Thein.

**Formal analysis:** Aye Sandar Zaw, Ei Shwe Sin Win, Alistair R. D. McLean, Frank M. Smithuis.

**Funding acquisition:** Frank M. Smithuis.

**Investigation:** Frank M. Smithuis.

**Methodology:** Aye Sandar Zaw, Nicholas J. White, Frank M. Smithuis.

**Project administration:** Frank M. Smithuis.

**Resources:** Frank M. Smithuis.

**Supervision:** Nicholas J. White, Frank M. Smithuis.

**Validation:** Aye Sandar Zaw, Ei Shwe Sin Win, Frank M. Smithuis.

**Visualization:** Aye Sandar Zaw, Soe Wai Yan, Frank M. Smithuis.

**Writing – original draft:** Aye Sandar Zaw, Vasundhara Verma, Nicholas J. White, Frank M. Smithuis.

**Writing – review & editing:** Aye Sandar Zaw, Ei Shwe Sin Win, Soe Wai Yan, Kyaw Sithu Thein, Vasundhara Verma, Alistair R. D. McLean, Thar Tun Kyaw, Nicholas J. White, Frank M. Smithuis.

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
