## [Editor Report · Decision Letter 0]

26 Jun 2023

Dear Dr Smithuis, 

Thank you for submitting your manuscript entitled "Successful elimination of falciparum malaria following the introduction of community-based health workers in Eastern Myanmar" for consideration by PLOS Medicine.

Your manuscript has now been evaluated by the PLOS Medicine editorial staff as well as by an academic editor with relevant expertise and I am writing to let you know that we would like to send your submission out for external peer review.

We also request at this time to include line numbers, to facilitate the review process. 

Please re-submit your manuscript within two working days, i.e. by Jun 28 2023 11:59PM.

Kind regards,

Katrien Janin, PhD

Senior Editor

PLOS Medicine

---

## [Decision Letter · Decision Letter 1]

30 Aug 2023

Dear Dr. Smithuis,

Thank you very much for submitting your manuscript "Successful elimination of falciparum malaria following the introduction of community-based health workers in Eastern Myanmar" (PMEDICINE-D-23-01742R1) for consideration at PLOS Medicine. 

[LINK]

In light of these reviews, we will not be able to accept the manuscript for publication in the journal in its current form, but we would like to consider a revised version that addresses the reviewers' and editors' comments. Obviously we cannot make any decision about publication until we have seen the revised manuscript and your response, and we plan to seek re-review by one or more of the reviewers. 

We expect to receive your revised manuscript by Sep 20 2023 11:59PM. Please email us (plosmedicine@plos.org) if you have any questions or concerns.

We look forward to receiving your revised manuscript. 

Sincerely,

Katrien Janin, PhD 

PLOS Medicine

plosmedicine.org

Comments from the academic editor:

The paper presents important findings and is clearly presented and written. A key issue to address in the revision, as raised by the reviewers, is that there was no control or comparison group because this is an analysis of the implementation of a intervention program. However, there remains the question of what would we have expected to happen to malaria incidence over that period in the absence of the intervention. Other factors (e.g. climate, rainfall, other malaria intervention activities) could have contributed to the reduction over time. It seems very unlikely that factors other than the CHW intervention program would have had a major effect on malaria reduction. The authors could include some comparison data to help understand that, such as malaria trends in neighboring regions, information about rainfall or other relevant climate factors over that time, any data on vectors. This should also be addressed further in the discussion to recap for the reader why we can conclude that the malaria reduction is wholly or mostly due to the intervention.

Your manuscript has been assessed by four reviewers whose reports can be found below. As you will see from the comments, the reviewers have raised a number of concerns that need addressing. Please carefully revise the manuscript to address all comments raised.

Causal language: given the study design please use associational language in the discussion and other sections. Please check and amend throughout. 

The polemic about the “expensive and time-consuming activities suggested by WHO” seems inappropriate. Please rephrase or remove.

For in-text reference, citations are placed within square parentheses and should precede punctuation. Please check and amend throughout. 

Please provide 95% CIs and p values for all results were appropriate, check and amend throughout.

For p values, please report these as p<0.001 and where higher as p=0.002 or p=0.050 (Please report all p values consistently to 3 decimal places - thousandths). Check and amend throughout.

Please specify the statistical test used to determine p value.

Suggest reporting statistical information as follows for clarity for the reader “…70%; 95%CI [65%, 75%]; Suggest use of commas as opposed to hyphens (as these can be confused with negative values) to separate upper and lower bounds. 

STUDY DESIGN:

Please ensure that the study is reported according to the STROBE guideline, and include the completed STROBE checklist as Supporting Information. Please add the following statement, or similar, to the Methods: ""This study is reported as per the Strengthening the Reporting of Observational Studies in Epidemiology (STROBE) guideline (S1 Checklist).""

When completing the checklist, please use section and paragraph numbers, rather than page numbers."

"Did your study have a prospective protocol or analysis plan? Please state this (either way) early in the Methods section.

c) In either case, changes in the analysis-- including those made in response to peer review comments-- should be identified as such in the Methods section of the paper, with rationale."

For all observational studies, in the manuscript text, please indicate: (1) the specific hypotheses you intended to test, (2) the analytical methods by which you planned to test them, (3) the analyses you actually performed, and (4) when reported analyses differ from those that were planned, transparent explanations for differences that affect the reliability of the study's results. If a reported analysis was performed based on an interesting but unanticipated pattern in the data, please be clear that the analysis was data-driven.

TITLE: 

Please revise your title according to PLOS Medicine's style. Please place the study design ("A randomized controlled trial," "A retrospective study," "A modelling study," etc.) in the subtitle (ie, after a colon).

DATA AVAILABILITY:

Please review the data availability statement. 

Can you please add ‘freely ‘ to your statement “The data underlying this publication are available upon request to the Mahidol-Oxford Tropical Medicine Research Unit Data Access Committee … “

ABSTRACT:

Abstract Background: 

The final sentence should clearly state the study aim/question.

Abstract Methods and Findings:

Please quantify the main results (with 95% CIs and p values).

Please check line 41: “.. examination rate was 1.33% (0.38–3.48%) per month.” Missing ‘95% CI’ ?

In the last sentence of the Abstract Methods and Findings section, please describe the main limitation(s) of the study's methodology.

Abstract Conclusions:

Please address the study implications without overreaching what can be concluded from the data; the phrase ""In this study, we observed ..."" may be useful.

Please interpret the study based on the results presented in the abstract, emphasizing what is new without overstating your conclusions.

Mention specific implications substantiated by the results.

AUTHORS SUMMARY:

At this stage, we ask that you include a short, non-technical Author Summary of your research to make findings accessible to a wide audience that includes both scientists and non-scientists. The Avalueuthor Summary should immediately follow the Abstract in your revised manuscript. This text is subject to editorial change and should be distinct from the scientific abstract. Please see our author guidelines for more information: https://journals.plos.org/plosmedicine/s/revising-your-manuscript#loc-author-summary

Ideally each sub-heading should contain 2-3 single sentence, concise bullet points containing the most salient points from your study.

In the final bullet point of ‘What Do These Findings Mean?’ Please include the main limitations of the study in non-technical language.

DISCUSSION:

Please present and organize the Discussion as follows: a short, clear summary of the article's findings; what the study adds to existing research and where and why the results may differ from previous research; strengths and limitations of the study; implications and next steps for research, clinical practice, and/or public policy; one-paragraph conclusion.

ACKNOWLEDGMENTS/ DECLARATIONS

Please remove all statements apart from acknowledgements, author contributions and abbreviations from the end of the main manuscript and include these only in the relevant parts of the manuscript submission form. Funding, competing interest, and data availability will be compiled as metadata.

REFERENCES:

Please use the "Vancouver" style for reference formatting, and see our website for other reference guidelines https://journals.plos.org/plosmedicine/s/submission-guidelines#loc-references

Please ensure that in the bibliography up to but no more than 6 author names are listed, followed by et al., in the event that more than 6 authors contribute to an individual study. Journal name abbreviations should be those found in the National Center for Biotechnology Information (NCBI) databases.

Please also ensure that any references to online-only sources include a date of accession, and in the reference list, please convert all italics to plain text.

FIGURES and TABLES

Please provide titles and legends for each individual table and figure

Figure 1: Please consider avoiding the use of red and green in order to make your figure more accessible to those with colour blindness

Table 1, 2 and 3: Please define the abbreviations.

SUPPORTING INFORMATION:

As above, please ensure that the study is reported according to the STROBE guideline, and include the completed STROBE checklist as Supporting Information. When completing the checklist, please use section and paragraph numbers, rather than page numbers. Please add the following statement, or similar, to the Methods: "This study is reported as per the Strengthening the Reporting of Observational Studies in Epidemiology (STROBE) guideline (S1 Checklist).

Comments from the reviewers:

Reviewer #1: This important retrospective analysis aims to assess the impact of introducing CHWs to deliver early diagnosis and treatment for malaria in hard-to-reach communities in Mon state, East Myanmar. 

Comments:

"The incidence of P. falciparum malaria (including mixed infections) declined by 70% (95% CI: 65-75%) each year of CHW operation. P. vivax malaria incidence declined by 56% (95% CI: 50-62%) per year. Malaria RDT positivity rates for P falciparum and P vivax declined by 69% (95% CI: 62-75%) and 53% (95% CI: 47-59%) per year respectively."

Can the authors please attempt to assess and comment on what the malaria rates were expected to do over time without any intervention in place?

"MAM introduced CHWs in remote communities in Mon state in consultation with the Myanmar National Malaria Control Programme (NMCP)."

and

"In collaboration with the NMCP, CHWs, and MAM mobile medical teams also engaged in mass distribution of long-lasting insecticidal nets (LLINs) during the first four years of the programme. Additional LLINs were made available for newcomers to the village, migrants, and pregnant women. In addition, CHWs and MAM mobile medical teams organized health education and community engagement discussions to inform the community and seek their feedback."

In addition to the above interventions, what other factors (i.e. confounders) may be affecting the trend seen?

"The introduction of CHWs providing community-based malaria diagnosis and treatment, integrated in basic health care services, in remote communities in Mon state led to a dramatic reduction in malaria."

Can the auhtors please consider whether this study design allows for causality to be inferred (i.e. 'led to')?

"RDT positivity rates and monthly blood examination rates were calculated from monthly malaria reports. Negative binomial mixed effects regression models were constructed with random intercepts and slopes to assess temporal changes in incidence and RDT positivity rates. Seasonal and geographic variations in outcomes were accounted for in the multivariate models. "

A technically appropriate and rigorous methodology has been applied.

"When modelling incidence and RDT positivity rates, the outcome was malaria (RDT-positive) results (per month, per CHW) and the primary exposure was years of CHW operation. The exposure variable for the incidence model was the population served by a CHW, and the exposure variable for the RDT positivity rate model was the number of RDTs performed."

Can the authors please clarify these models? For instance:

When modelling incidence, is it assumed that all incident cases have been identified in the population served by a CHW?

When modelling RDT positivity rates, does the exposure variable assume a representative sample (i.e. such that the rate can be generalised)?

Figure 4: Can CIs for estimates please be presented on these graphs?

Reviewer #2: Thanks for this nice documentation of malaria elimination in Mon. The story that routine diagnosis and treatment likely played a critical role in eliminating malaria there, and is likely both more effective and more sustainable than aggressive reactive approaches, is an important one. A few suggestions to strengthen it:

1) L56-58: "The World Health Organization has developed several detailed 'malaria elimination manuals' in recent years". The five references here do not actually include what I believe is the most current WHO elimination recommendations, a document called A Framework for Malaria Elimination from 2017: https://www.who.int/publications-detail-redirect/9789241511988. The framework gets rid of the ideas of discrete "phases" and instead describes elimination on a spectrum, and it rejects the idea that malaria operations should be based on the arbitrary thresholds described in this paragraph, such as shifting to different types of programs when there is 1 case per 1,000 per year - such decisions are highly context specific. The main content here - that elimination is recommended to involve targeting strong activities to foci of transmission - remains largely unchanged, but I'd suggest updating the framing.

2) L115 - PCD of course more specifically means that febrile individuals had to seek treatment from the CHW - they would not actively seek out fevers in the community. Can you confirm this is correct? Later there is a description of how the CHWs also provided ACD, but those data are presented separately?

3) L126 - Is there any data on adherence to the 8 weeks of PQ?

4) I do not see any evaluation on reporting rates or how they varied over time - it might be reassuring to show those if available as evidence that the observed declines are reliable.

5) It would be interesting to know the effect of the integration of other services in 2013-2014 - e.g., did this affect testing rates?

6) Ascribing causality is challenging because this model shows only that malaria declined over time - there is no control or pre-post comparison. If malaria were declining unrelated to CHWs (which I do not believe!), would you not find a similar result? If there are sufficient data from before substantial uptake of CHW services, I would consider trying more of an interrupted time series to show the trend in malaria in places where CHWs were not yet introduced (or where uptake was low) and then the change following introduction (or increase in uptake).

7) In addition, the timing of the LLIN distributions is not specified, but obviously that could be having a substantial effect in addition to the effect of treatment. Conclusions like "this tipping point was readily reached with this single intervention" don't seem supported by the fact that there were several interventions mentioned, including routine case management + ACD + LLINs. Analysis more clearly showing those latter approaches had minimal effect would be helpful. 

Reviewer #3: The MS entitled "Successful elimination of falciparum malaria following the introduction of community based health workers in Eastern Myanmar" is an excellent work that most certainly deserves and needs to be published. The conclusions reached in the MS are of utmost importance for refining public health malaria control measures and most importantly, elimination strategies in Greater Mekong Sub-region, especially in hard-to- reach and resource-poor areas, which often also are the highest malaria risk areas. The MS is well written, the data have been appropriately analysed and clearly presented.

Reviewer #4: Successful elimination of falciparum malaria following the introduction of

4 community-based health workers in Eastern Myanmar

The manuscript is well written. It highlights that improved access to timely and effective treatment and vector control by community health workers can impact the malaria burden. The authors report elimination of P. falciparum malaria and a significant decline in P. vivax in the study area. The following few issues should be addressed:

1. Malaria burden data after the project is handed over to NMCP will provide information on the sustainability of the work.

2. Line 123: "To reduce transmissibility, a single dose of primaquine (0.75mg base/kg) was..." for falciparum cases. The recommended dose of primaquine as an antigametocidal is 0.25mg/kg. Why was there three times this dose used? Perhaps it is a typo error.

3. In the "Additional intervention" section, the mass distribution of LLINs and additional LLINs to new arrivals, migrants, and pregnant women in the first 4 years was given. This is an important measure. The author presented the number of LLINs distributed in the results. The introduction of LLINs should also be presented in the disease burden figure with the rainfall. In addition, the authors have ignored the potential impact of this intervention and should discuss it in the discussion section and conclusion.

[LINK]

---

## [Decision Letter · Decision Letter 2]

27 Oct 2023

Dear Dr. Smithuis,

Thank you very much for re-submitting your manuscript "Successful elimination of falciparum malaria following the introduction of community-based health workers in Eastern Myanmar: A retrospective analysis." (PMEDICINE-D-23-01742R2) for review by PLOS Medicine.

I am pleased to say that we are planning to accept the paper for publication in our journal after minor revisions. The remaining comments that we would like you to address are listed below. 

We expect to receive your revised manuscript within 1 week. Please email us (plosmedicine@plos.org) if you have any questions or concerns, or like an extension. 

If you have any questions in the meantime, please contact me (kjanin@plos.org) or the journal staff on plosmedicine@plos.org.  

We look forward to receiving the revised manuscript by Nov 03 2023 11:59PM.   

Sincerely,

Katrien Janin, PhD

Senior Editor 

PLOS Medicine

plosmedicine.org

Requests from Editors:

In line with reviewer 1, we ask you to include some additional text from their point-by-point response into the paper.

We also feel that language around the WHO recommendations is still quite strong, but we do appreciate the importance of the raised point and therefore are inclined to let this stand.

We also invite you to provide us with any X (formerly known as Twitter) handle(s) that would be appropriate to tag, including your own, your co-authors’, your institution, funder, or lab. Please respond to this email with any handles you wish to be included when we post about this paper.

Comments from Reviewers:

Reviewer #1: Many thanks to the authors for responding to each comment in turn. 

Can the following responses to reviewer please be further incorporated within the manuscript?

Author responses:

"This is a good question but it is hard to answer. It is difficult to know what the malaria rates were and

would be over time, because malaria data in Myanmar are notoriously unreliable, in particular in remote

communities. In most remote communities there were no official health staff and fever patients were

managed by unofficial health providers (locally called "quacks") who usually did not have/use

diagnostics, and treated malaria - often delayed - outside official guidelines. They did not report

malaria.

As malaria transmission is low to moderate in this region, and there have been no major climatic or

environmental changes, we would expect that malaria transmission would probably continue on a

similar level as before without our intervention. The increasing availability of artemisinin treatment in

the market, and therefore also in the hands of the quacks, might have reduced transmission somewhat

over the past decades but we cannot be sure.

Only when we introduced CHWs, quality diagnosis and treatment was easy available and data were

collected regularly and systematically. The uptake was high because the services were in the village and

free of charge."

and

"Besides the introduction of CHWs with RDTs, antimalarials and LLINs, there were no substantial antimalaria interventions. The communities are very small and we would surely have noticed other

substantial interventions.

It is difficult to imagine other factors that could have influenced such a large number of villages over a

distance of 250 km, except for climate factors. However, the project period was short and there were no

substantial climate changes."

and

"Yes-the CHW were the only source of immediate health care and the uptake of their services is high. We

cannot exclude self treatment but there is no reason to believe either that this was common or that it

changed during the period of observation."

and

"Yes - the samples were taken from the entire population and the proportions were of all cases seen by

the CHWs.

The exposure variable is RDTs performed on patients who sought the care of a CHW for their fever. As

mentioned above, the uptake of CHW services - and RDTs performed - was high throughout the study

period. Whether the RDT positivity rates of febrile individuals who sought the care of a CHW are

representative of all febrile individuals (including those who do not seek the care of a CHW) is not

something we are able to test, but we think it is a reasonable assumption. "

Reviewer #2: Thanks for addressing my comments - it is a shame there is no useful comparison group that would allow for more rigorous evaluation of the effect of the CHWs, but I understand the context and I think this is still a very useful contribution regardless. I have no further critiques.

Reviewer #4: The authors responded to the queries I raised satisfactory.

---

## [Editor Report · Decision Letter 3]

3 Nov 2023

Dear Dr Smithuis, 

On behalf of my colleagues and the Academic Editor, I am pleased to inform you that we have agreed to publish your manuscript "Successful elimination of falciparum malaria following the introduction of community-based health workers in Eastern Myanmar: A retrospective analysis." (PMEDICINE-D-23-01742R3) in PLOS Medicine.

Best wishes, 

Katrien G. Janin, PhD 

Senior Editor 

PLOS Medicine